# Immunogenicity of an Inactivated *Senecavirus A* Vaccine with a Contemporary Brazilian Strain in Mice

**DOI:** 10.3390/vaccines12080845

**Published:** 2024-07-26

**Authors:** Amanda de Oliveira Barbosa, Danielle Gava, Caroline Tochetto, Leonardo Clasen Ribeiro, Ana Paula Almeida Bastos, Marcos Antônio Zanella Morés, Rejane Schaefer, Marcelo de Lima

**Affiliations:** 1Laboratório de Virologia e Imunologia, Universidade Federal de Pelotas, Capão do Leão 96160-000, RS, Brazil; barbosa.oamanda@gmail.com (A.d.O.B.); leonardo.clasen@gmail.com (L.C.R.); 2Embrapa Suínos e Aves, BR 153, Km 110, Distrito de Tamanduá, Concordia 89715-899, SC, Brazil; daniellegava@gmail.com (D.G.); caroline.tochetto@colaborador.embrapa.br (C.T.); ana.bastos@embrapa.br (A.P.A.B.); marcos.mores@embrapa.br (M.A.Z.M.); rejane.schaefer@embrapa.br (R.S.)

**Keywords:** *Senecavirus A*, inactivated vaccine, representative strain

## Abstract

*Senecavirus A* (SVA) is a picornavirus that is endemic in swine, causing a vesicular disease clinically indistinguishable from other vesicular diseases, like foot-and-mouth disease. The widespread viral circulation, constant evolution, and economic losses caused to the swine industry emphasize the need for measures to control the agent. In this study, we evaluated the immunogenicity of a whole-virus-inactivated vaccine using a representative contemporary Brazilian SVA strain in Balb/ByJ mice. The animals were vaccinated with two doses by an intramuscular route. The humoral response induced by the vaccination was evaluated by an in-house ELISA assay for IgG detection. The cellular response was assessed by flow cytometry after in vitro SVA stimulation in splenocyte cultures from vaccinated and non-vaccinated groups. Protection against SVA was assessed in the experimental groups following an oral challenge with the homologous virus. The vaccination induced high levels of IgG antibodies and the proliferation of CD45R/B220^+^sIgM^+^, CD3e^+^CD69^+^, and CD3e^+^CD4^+^CD44^+^CD62L^−^ cells. These results indicate the immunogenicity and safety of the vaccine formulation in a murine model and the induction of humoral and cellular response against SVA.

## 1. Introduction

*Senecavirus A* (SVA) is a non-enveloped virus characterized by a positive-sense RNA genome with approximately 7 kb. The VP1, VP2, VP3, and VP4 proteins assemble into an icosahedral structure to form the viral capsid (~30 nm) [1]. The SVA genome encodes a unique open reading frame (ORF), which follows the organization of all picornaviruses: L-4-3-4 (leader; four polypeptides from P1; three polypeptides from P2; four polypeptides from P3) [2]. Initially, the ORF encodes a polyprotein that is proteolytically cleaved, resulting in twelve proteins comprising four structural and seven non-structural proteins [1].

SVA was discovered in 2002 as a contaminant on PER.C6 cells (human embryonic retinal cells) [3]; however, only in 2007 was it associated with vesicular disease in pigs in Canada [4]. The virus gained relevance as a swine pathogen after 2014–2015 when outbreaks of vesicular disease associated with SVA were reported in many countries, including Brazil [5,6]. SVA infection in pigs causes vesicles on the oral mucosa, snout, coronary bands, and hooves [7]. Currently, SVA is disseminated worldwide, resulting in significant economic losses due to the resemblance of lesions with other vesicular diseases [5], mainly foot-and-mouth disease (FMD), a disease officially listed by the World Organization for Animal Health.

The phylogenetic analysis of SVA strains revealed 6.32% genetic nucleotide divergence among historical (detected between 1988 and 2010) and contemporary (post-2010) SVA strains [8]. Notably, contemporary strains display more virulence than their historical counterparts [9]. Thus, the continuous circulation, dissemination, and ongoing evolution of SVA has led to an urgent call for effective vaccines. Zeng et al. (2023) have identified genetic modifications among SVA isolates that could result in antigenic variation and vaccine failures [10]. Experimental studies have tested different types of vaccines in pigs, including attenuated [11], inactivated [12,13,14,15], virus-like particles [16,17], and subunit [18] vaccines. All these vaccines protected pigs from clinical disease.

Even though several vaccine trials for SVA have been conducted directly in pigs, the use of mice as an experimental model for the evaluation of vaccines for SVA has already been described [17,19,20]. In pre-clinical investigations, this approach may be more cost-effective than directly testing the target species. The mouse model represents a faster and more accessible assessment of the immunogenicity and protection induced by a vaccine [21]. This study aimed to evaluate the immunological profile of mice vaccinated with an inactivated vaccine against SVA using a representative contemporary Brazilian isolate.

## 2. Materials and Methods

### 2.1. Viral Amplification and Sequencing

Fifty SVA isolates, provided by the official Brazilian Veterinary Laboratory from the Ministry of Agriculture, Livestock and Food Supply (LFDA/MG-MAPA), were selected considering their geographic location and temporal distribution. The viruses were from pigs showing vesicular disease from farms located in eight Brazilian states from 2018 to 2021. To achieve viral recovery, these samples were inoculated in H1299 cells (human non-small cell lung carcinoma) (ATCC CRL-5803) with an RPMI-1640 medium (Sigma-Aldrich^®^, St. Louis, MO, USA) supplemented with 10% Fetal Bovine Serum (FBS, Gibco, Waltham, MA, USA), 2 mM L-Glutamine, Penicillin (100 U/mL), and Streptomycin (100 µg/mL) (Sigma-Aldrich^®^) at 37 °C with 5% CO_2_ for 3 days.

Viral RNA was extracted from 200 µL of the cell supernatant using the BioGene DNA/RNA viral kit (Bioclin Quibasa, Belo Horizonte, Brazil), according to the manufacturer’s instructions. The cDNA synthesis was performed with the iScript™ cDNA Synthesis Kit (Bio-Rad, Hercules, CA, USA) in a 20 µL reaction using random primers, following the manufacturer’s instructions. A primer-walking RT-PCR sequencing approach was used for sequencing the full genome of the SVA isolates [22]. For this, nine sets of overlapping primers were used along with the GoTaq^®^ Colorless Master Mix (Promega, Madison, WI, USA). The RT-PCR products were subjected to electrophoresis in 1% agarose gel and purified using the Wizard SV gel and PCR clean-up System (Promega) according to the manufacturer’s instructions. The purified products were quantified in the Qubit 3.0 fluorometer using the Qubit dsDNA BR Assay Kit (Life Technologies, Carlsbad, CA, USA) according to the manufacturer’s protocols. Sequencing was performed on the ABI 3130xl Genetic Analyser using the BigDye Terminator kit v. 3.1 (Applied Biosystems, Waltham, MA, USA). The reads were assembled using the phred/phrap/consed package [23]. All consensus bases have quality with a Phred value of at least 20. The completeness of assembled genomes was verified using BLASTn. The single ORF of SVA was determined using ORFinder (NCBI) and checked with BLASTx.

### 2.2. Selection of the Candidate Vaccine Strain

The selection of the SVA strain for the vaccine formulation was carried out using the PARNAS algorithm [24]. For this purpose, a dataset containing the 50 Brazilian sequences generated in this study, along with all SVA genome sequences available in GenBank (NCBI) up to 2022, was built. Nucleotide alignment was generated using MAFFT v7.490 [25] and manually corrected with Aliview v1.28 [26]. Final multiple sequence alignment included the coding region (~6546 in length) of 328 SVA collected from pigs globally, including 64 sequences from Brazil. To infer the evolutionary relationships for this dataset, a maximum likelihood (ML) phylogenetic tree was reconstructed using the IQTREE v2.1.2, following the automatic best-fit model selection process (GTR + F + I + G4). The statistical support for branches within the inferred tree was assessed using the SH-like approximate Likelihood-Ratio Test (SH-aLRT) [27] and Ultrafast Bootstrap approximation (UFBoot) approximation with 1000 pseudoreplicates [28,29]. Based on this global phylogenetic inference (Appendix A), the more recent sequences of SVA strains isolated in Brazil (2018–2021) (*n* = 51) were used to infer a separate ML tree using the settings described previously. Additionally, a multiple amino acid alignment (ORF1, 2181 aa) was generated using MAFFT. A PARNAS algorithm was executed with the option to rescale the tree edges with the number of ORF1 amino acid substitutions and with 2% amino acid divergence between SVA sequences. The pairwise amino acid identity matrix was generated using SDT v1.2 (Appendix A).

### 2.3. Vaccine Formulation

The most representative SVA isolate (SVA/BRA/GO/946-22/2019) Genbank accession number OR567056; CMISEA ID 3258; SISGEN ID AFFB5D4) was amplified and titrated in H1299 cells, as described above. The viral titer used in the vaccine formulation was 10^7^ TCID_50_/100 µL.

For viral inactivation, 1.5% binary ethylenimine (BEI) (Sigma-Aldrich^®^) was added to the filtered and previously titrated viral suspension. The mixture was kept under agitation at 37 °C for 8 h, as described by Bahnemann (1990) [30]. To assess the vaccine safety, the viral suspension was seeded on a MacConkey agar (OXOID Ltd., Nepean, ON, Canada) and Blood agar base and incubated at 37 °C for 24 h, and we inoculated the inactivated viral suspension into H1299 cells at 37 °C with 5% CO_2_ for 5 days. The inactivated viral suspension was emulsified with the adjuvant MONTANIDE™ ISA 201 VG (Seppic, Courbevoie, France) at 55:45 (*v*:*v*; virus: adjuvant) at 35 °C, according to the manufacturer’s recommendations.

### 2.4. Animals and Vaccine Protocol

The immunogenicity and protective efficacy of the inactivated vaccine for SVA were evaluated at Embrapa Swine and Poultry in compliance with the Animal Use Ethics Committee (protocol number 15/2021). For the vaccine trial, forty (40) 6–8-week-old female Balb/cByJ, raised in specific pathogen-free conditions, were randomly divided into four groups (Table 1).

The animals received two doses of vaccine by an intramuscular route (10^7^ TCID_50_/100 µL + ISA 201 VG, IM), 15 days apart (Figure 1). The first dose was applied on day 0 (zero). Fecal samples were collected from grouped-housed mice. For this, mice were transferred to another box, and fresh feces were obtained. Blood samples were collected individually through a retro-orbital puncture under sedation. Feces and blood were taken for all groups on days 0, 15, 36, and 41 after immunization or RPMI administration. The mice received 200 µL of saline solution after blood collection, infused intraperitoneally. At 36 dpv, mice from non-vaccinated (GII) and vaccinated (GIV) groups were challenged with 100 µL of the homologous virus (2 × 10^7^ TCID_50_/100 µL) orally in a laboratory biosafety level 3 (BSL-3) where they remained until euthanasia at 41 dpv. After the challenge, fecal samples from challenged groups (GII and GIV) were collected daily in a pool per group. To carry out the vaccination and blood collection, mice were sedated with 0.02 mL of a solution prepared with ketamine (10%, 60 mg/kg) (Syntec) and xylazine (2%, 10 mg/kg) (CEVA) administered intraperitoneally. The animals were monitored daily throughout the experiment. At 41 dpv, all mice were euthanized with an intraperitoneal ketamine (300 μg/g) and xylazine (40 μg/g body weight) injection. The hearts, lungs, livers, kidneys, and duodenums were harvested from each group for subsequent histopathological assessment and to determine the viral load in the tissues following the challenge (GII and GIV groups). Spleens obtained from non-vaccinated (GI) and vaccinated (GIII) in vitro stimulated groups were collected and processed to isolate white blood cells. These cells were utilized for the in vitro stimulation, following the procedure described below.

### 2.5. RNA Extraction and RT-qPCR

To evaluate the viral load in feces (all groups) and tissues (challenged groups: GII and GIV), each biological sample was weighted to standardize the volume for RNA extraction. Afterward, feces and tissues were minced using a sterile scalpel, resuspended in 0.5 mL of PBS (phosphate-buffered saline), and homogenized in a vortex for 15 s. Then, the suspension was centrifuged (3000 rpm, 5 min, 4 °C), and the supernatant was used for RNA extraction. Nucleic acid was extracted from pooled fecal samples and from the hearts, lungs, spleens, livers, kidneys, and duodenums using a MagMAX™ Viral RNA Isolation kit (Thermo Fisher Scientific, Waltham, MA, USA), following the manufacturer’s instructions. The detection of SVA RNA in samples was evaluated using an RT-qPCR assay [22]. Cq (cycle of quantification) values greater than 38 were considered negative.

### 2.6. Morphologic Assessment

Hearts, lungs, livers, kidneys, and duodenums were harvested from mice on 41 dpv to investigate the presence of histopathological lesions. The mice received an anesthetic overdose to euthanize. Tissues were fixed in 4% buffered paraformaldehyde, embedded in paraffin, cut into sections that were 4 µm thick, and then stained with hematoxylin and eosin (H&E). The morphologic evaluations were conducted blindly by a pathologist.

### 2.7. ELISA IgG for SVA Antibodies

Serum samples collected on days 0, 15, 36, and 41 dpv were subjected to an in-house ELISA test for IgG detection to assess the humoral immune response induced by vaccination. For this, 96-well plates (ref. 9017, Corning^®^, New York, NY, USA) were coated with a 100 µL/well containing 37.8 µg/mL of the homologous vaccine virus in a carbonate–bicarbonate buffer (0.05 M and pH 9.6) and incubated overnight at 4 °C.

Subsequently, serum samples were diluted 1:20 for IgG detection in a dilution solution (PBS, 0.05% Tween 20, and 1% Bovine Serum Albumin Fraction V, Roche, Basel, Switzerland), and 100 µL was added in each well. Next, the plates were incubated with 100 µL/well of Rabbit Anti-Mouse IgG(H + L)-HRP (SouthernBiotech, Birmingham, BL, USA) at a dilution of 1:5000. All incubations were performed in a humidified chamber at 37 °C for 1 h. After each step, plates were washed three times using the wash solution (PBS + 0.05% Tween 20). Finally, 100 µL/well of TMB (3,3′,5,5′-tetramethylbenzidine, Sigma-Aldrich^®^) was added for 15 min to analyze antibody binding. Then, the reaction was stopped by adding 50 µL/well of sulfuric acid (H_2_SO_4_, 2 M). The absorbance was determined in a spectrophotometer at an optical density (OD) of 450 nm. The OD values of sera collected on day 0 were used to calculate the cutoff for IgG ELISA, as described by Frey, Di Canzio, and Zurakowski (1998) [31].

### 2.8. In Vitro Stimulation and Flow Cytometry

To evaluate the cellular immune response, the mice’s spleens from in vitro stimulated groups (GI: non-vaccinated and GIII: vaccinated) were collected in an RPMI 1640 medium at necropsy. Afterward, each spleen was mechanically dissociated and filtered through a 70 µm filter (Nylon Cell Strainer, Corning). The tissue suspension was centrifuged (1100 rpm, 5 min), and the cells were resuspended in ammonium chloride lysis solution (ACK) and incubated for 3 min to lyse the red blood cells. The lysis reaction was stopped by adding the RPMI 1640 medium with 2% FBS. The cells were washed twice and cryopreserved at −196 °C in a solution of 95% FBS + 5% Dimethyl sulfoxide (DMSO, Sigma-Aldrich^®^).

Following a rapid thawing process, the number of viable cells was determined by counting with trypan blue (0.4%) in a Neubauer chamber. Then, the cells were resuspended in DPBS (Dulbecco’s Phosphate Buffered Saline) and labeled with 2.5 µM carboxyfluorescein succinimidyl ester (CFSE) by applying the CellTrace™ CFSE Cell Proliferation kit (Invitrogen, Waltham, MA, USA). The incubation protocol was carried out according to Fonseca et al. (2023) [32]. Next, the cells were resuspended in the RPMI 1640 medium, and 1 × 10^6^ cells/well were plated in 24-well plates (3 wells per mouse). The cells were individually stimulated with 8000 TCID_50_ of SVA, using the same virus employed in the vaccine formulation. For positive control, 3 µg/mL Concanavalin A (Sigma-Aldrich^®^) was used, and for negative control, the RPMI 1640 medium was added. The lymphocytes were cultured with the addition of 2 mM l-glutamine (Gibco), 500 µM 2-mercaptoethanol (Sigma-Aldrich), 10% heat-inactivated FBS, and 1% penicillin/streptomycin (Sigma-Aldrich^®^) in a final concentration of 1 × 10^6^ cells/mL. The cells were incubated at 37 °C with 5% CO_2_ and were analyzed after 72 h of cell culturing on a Cytoflex S cytometer.

To assess the proliferation and activation status of cell subpopulations (T lymphocyte subset, activated T lymphocytes, naive and memory T cells, and B lymphocyte subset), the stimulated cells were distributed in 96-well plates and incubated with monoclonal antibodies (mAbs) (Table 2) for 50 min at 37 °C and 5% CO_2_. The concentrations of antibodies used followed the recommendations of the respective manufacturers. Fifty thousand events were read per well in the Cytoflex S cytometer (Beckman Coulter Life Sciences, Brea, CA, USA), and subsequent analyses were conducted in the FlowJo software (New V9/10) (Becton-Dickinson, Franklin Lakes, NJ, USA). Positive cells were identified through standard gating procedures utilizing Fluorescence Minus One (FMO) controls. The cells were labeled with 0.25 µg/10^6^ cells of 7-amino actinomycin D (7-AAD, eBioscience, San Diego, CA, USA) and then kept in the incubator for 30 min before being analyzed on the cytometer. Cells that tested positive for 7-AAD were classified as dead.

### 2.9. Statistical Analysis

RT-qPCR and ELISA data were used for descriptive analyses. RT-qPCR results were presented by the group, and humoral response data (ELISA for IgG) were shown individually with the mean ± standard deviation. The two-sample Student’s *t*-test was conducted to assess differences in the ELISA data between the GI + GII (non-vaccinated) and GIII + GIV (vaccinated) groups at 15, 21, and 36 dpv (before challenge) and between the GI (non-vaccinated) and GIII (vaccinated) groups at 41 dpv. The one-sample Student’s *t*-test was conducted to assess differences between the in vitro stimulated GI (non-vaccinated) and GIII (vaccinated) groups in flow cytometry data. *p* values ≤ 0.05 were considered statistically significant. All analyses and graphs were performed using GraphPad Prism (version 7.0) software (GraphPad Software Inc., San Diego, CA, USA).

## 3. Results

### 3.1. Selection of a Vaccine Candidate for SVA

The complete coding sequence (CDS) of the 50 SVA isolates was obtained and deposited in Genbank under accession numbers OR567038-OR567087; CMISEA ID 3252–3353; SISGEN ID AFFB5D4. The selection of a vaccine candidate strain was considered after in silico analyses that included all sequences of Brazilian SVA isolated from 2018 to 2021. The isolate SVA/BRA/GO/946-22/2019 (Genbank accession number OR567056; CMISEA ID 3258; SISGEN ID AFFB5D4) was chosen as the best representative sequence that covered 98% of the amino acid genetic diversity observed in the Brazilian SVA sequences. The percent of amino acid identity among the 51 Brazilian SVA isolates was 98.2% or higher. The sequence selected by Parnas as the best representative showed an overall amino acid identity ranging from 98.67% to 99.82% compared to the other sequences generated in this study.

### 3.2. Vaccine Safety

There was no bacterial growth or viral replication in the inactivated viral suspension (Appendix A). Following each administration of the vaccine, the animals were carefully monitored for any potential adverse responses. No changes in animal behavior patterns or adverse reactions, whether local or systemic, to the vaccine formulation were observed. The vaccine was shown to be safe.

### 3.3. Viral Load in Feces and Tissues

There was no detection of SVA RNA in the feces collected during the experiment before the challenge (Figure 2). After the challenge, SVA RNA was detected in feces collected from the challenged groups (GII and GIV). The peak of viral shedding occurred on day one post-challenge (dpc) (Cq 27.2–31.6) and decreased until day five when there was no further detection. Apart from days one and four dpc, the viral shedding pattern was similar between the non-vaccinated (GII) and vaccinated (GIV) groups. At one and four dpc, viral shedding was higher in the vaccinated group (GIV) compared to the non-vaccinated group (GII). Furthermore, SVA RNA was also detected in tissues collected from two animals from the non-vaccinated group and challenged (GII) by RT-qPCR. In one animal, viral RNA was detected in the lungs (Cq 37.4) and heart (Cq 37.3), while in the other, viral RNA was detected in the lungs (Cq 37), liver (Cq 33), spleen (Cq 33.6), kidneys (Cq 36), and duodenum (Cq 35.1). There was no viral RNA detection in mice’s organs in the vaccinated and challenged group (GIV).

### 3.4. Evaluation of Microscopic Lesions in Tissues

No lesion suggestive of SVA infection was observed in the histopathological evaluation. Nonspecific findings were noticed in the organs of some animals from all groups, such as areas with hypereosinophilia or vacuolization of some muscle fibers in the myocardium (8/40), active Kupffer cells in the liver (12/40), mild interstitial lymphoid infiltration in the kidneys (10/40), and slight infiltration of plasma cells in the lamina propria of the duodenum (34/40).

### 3.5. Immunogenic Potential of Vaccine

Considering the established cutoff (OD of 0.225) for the in-house IgG ELISA, antibodies were detected in mice after vaccination at 15 and 36 dpv (*p* < 0.0001) and after in vitro stimulation at 41 dpv (*p* < 0.0001) (Figure 3). The mean OD of vaccinated groups (GIII + GIV) was 2.85 times higher than non-vaccinated groups (GI + GII) on day 15 and 8.26 times higher on day 36, indicating seroconversion in the vaccinated groups. Comparing the data obtained on day 15 (referring to the first dose) and day 36 (referring to the complete vaccination protocol), the OD value on day 36 was 2.79 times higher, indicating anamnestic serological responses. None of the mice from non-vaccinated groups (GI and GII) were considered positive for SVA during the experimental days by the in-house IgG ELISA.

### 3.6. Evaluation of the Cellular Response against the SVA Vaccine

To evaluate the cell proliferation profile, splenocytes obtained from groups GI (non-vaccinated) and GIII (vaccinated) were in vitro stimulated and subjected to flow cytometry. The gates were determined as previously described [32] (Appendix A). Dead cells were excluded from the analysis by labeling them with 7-AAD, and the lymphocyte proliferation following in vitro stimulation was assessed by CFSE. Through the specific labeling, the proliferation of the following lymphocyte subsets was observed: CD3e^+^CD4^+^CD8α^+^ (double-positive), CD3e^+^CD4^+^ (CD4^+^ T lymphocytes), CD3e^+^CD8α^+^ (CD8^+^ T lymphocytes), CD3e^+^CD69^+^ (very early activation T cell), CD3e^+^CD69^+^CD25^+^ (effector T cells), CD3e^+^CD4^+^CD44^+^CD62L^+^ (central memory CD4^+^ T lymphocytes), CD3e^+^CD4^+^CD44^+^CD62L^−^ (effector memory CD4^+^ T lymphocytes), CD45R/B220^+^sIgM^+^ (immature and mature B cells or transitional B cells), and CD45R/B220^+^sIgM^+^CD23^+^ (mature resting conventional B cells). There was no significant increase in proliferation observed on the panel of the T lymphocyte subset (CD3e^+^CD4^+^CD8α^+^, CD3e^+^CD8α^+^, CD3e^+^CD4^+^). In the remaining panels, the fold change over the non-vaccinated group was 2.26 for immature and mature B cells or transitional B cells (B220^+^IgM^+^) (*p* = 0.0407), 2.17 for very early activation of the T cell (CD3e^+^CD69^+^) (*p* = 0.0025), and 1.88 for effector memory CD4^+^ T lymphocytes (CD3e^+^CD4^+^CD44^+^CD62L^−^) (*p* = 0.0001) (Figure 4).

## 4. Discussion

*Senecavirus A* (SVA) has become endemic in the pig population in several countries and has a significant economic impact, mainly due to its clinical resemblance to FMD [5]. The development of vaccines is crucial to mitigate the negative effects caused by SVA infection, including losses in herd productivity and the swine industry. The most effective vaccines against other picornaviruses, such as FMD in cattle and poliovirus in humans, rely on either inactivated viruses or attenuated forms of the virus [33,34]. Inactivated whole virus vaccines have several advantages, including the stimulation of humoral immune responses against multiple targets, antigens, or epitopes. Since these vaccines are based on inactivated virus particles, there is no viral replication in the host, resulting in the absence of clinical disease and reversion to virulence. These inactivated whole virus vaccines are also cost-effective and comparatively simple to produce [35]. In this study, an inactivated vaccine against SVA using a representative contemporary Brazilian isolate was immunogenic in a murine model.

Murine models are widely utilized in pre-clinical evaluations due to their consistent genetic background, easy handling, and cost-effectiveness [21]. They have also been used to evaluate vaccine platforms targeting SVA [17,19,20]. Previous studies have indicated the susceptibility of mice to SVA infection [3,22]. In our study, an anamnestic serological response triggered by the inactivated vaccine was observed. Animals from the vaccinated groups (GIII and GIV) developed a robust IgG antibody response against SVA with an increase of 8.26 times in the OD mean after the booster compared to the non-vaccinated groups (GI and GII). This result highlights the importance of a second vaccine dose in enhancing the humoral immune response, a recommended practice for inactivated vaccines [36]. However, in a study assessing an inactivated vaccine for SVA in Balb/c mice, Li et al. (2022) administered only a single vaccine dose [19]. Their findings revealed the presence of neutralizing antibodies, reaching the highest titer of 1:128 at 28 days after vaccination. In another study evaluating a VLP vaccine for SVA in C57BL/6 mice, Zhang et al. (2023) noted the induction of neutralizing antibodies at 35 days after vaccination [17]. In pigs, IgG antibodies can be detected for at least 60 days after SVA infection. IgG antibody response is responsible for late viral neutralization, contributing to the protection of pigs against reinfection [37]. Considering the important role of neutralizing antibodies in SVA infection, sera from SVA-vaccinated mice were evaluated in a serum neutralization assay. However, sera exhibited toxicity to H1299 and BHK-21 cells. Upon reaching the 1:64 dilution, cytotoxicity was no longer observed; however, no detection of neutralizing antibodies was observed. Despite the methodological limitation observed in our study, we could not rule out the presence of neutralizing antibodies in vaccinated animals.

To assess the protection against SVA induced by the vaccination, a challenge with a substantial viral load of 2 × 10^7^ TCID_50_/100 µL was administered orally to the challenged groups (GII—non-vaccinated and GIV—vaccinated). Five days post-challenge, a minimal viral load was observed in the organs, with only two animals from the non-vaccinated and challenged group (GII) showing detectable SVA RNA levels. This contrasts with the findings of Li et al. (2022), in which viral RNA was identified in several organs of mice at 14 days post-challenge [19]. Additionally, the viral load in the fecal samples collected from the two challenged groups (GII—non-vaccinated and GIV—vaccinated) was detectable only up to 4 days post-challenge. Conversely, studies in pigs have shown that SVA shedding in feces can persist for as long as 28 days after infection [7]. Moreover, the viral RNA peak occurred on the first day post-challenge. We hypothesized that this peak could be attributed to the oral content employed in the challenge, which was promptly shed in the feces without establishing an infection. In line with this observation, the histopathological findings consistently revealed nonspecific lesions in all groups without any association with SVA infection. Overall, these findings suggest that SVA did not establish a productive infection in most of the mice used in our study. The absence of a productive infection in the challenged mice prevented us from assessing the vaccine’s ability to reduce infection or prevent disease in the murine model, thereby hindering the analysis of vaccine effectiveness. Nonetheless, the generation of SVA-specific antibodies enabled us to evaluate the vaccine’s immunogenicity.

The proliferation of the immune cells from the spleen, induced by vaccination, was evaluated after the in vitro stimulation. Regarding B cells, a notable increase in CD45R/B220^+^sIgM^+^cells (B-1 cells) in the GIII group (vaccinated and in vitro stimulated) over the GI group (non-vaccinated and in vitro stimulated) was verified, suggesting the induction of a humoral immune response by vaccination, which was also observed in ELISA assays. B-1 cells are known for their elevated expression of surface immunoglobulin M (sIgM) and the B cell isoform of CD45R/B220 [38]. B1 cells are a specific type of B cell lymphocytes that have a role in the humoral immune response [39]. Although the immune response consisted of increased B-1 cell-derived IgM (CD45R/B220^+^sIgM^+^) production in the spleen, low IgM levels in serum were observed throughout the study. Nevertheless, this circumstance does not seem to have interfered with the maturation and class-switching of B cells, as elevated levels of IgG were detected in the serum of vaccinated animals. Studies have shown that B-1 cells not only produce antibodies but also play an active role in immune responses triggered by pathogens.

Regarding T cells, our findings showed a notable proliferation of the CD3e^+^CD69^+^ and CD3e^+^CD4^+^CD44^+^CD62L^−^ subgroups in the GIII group (vaccinated and in vitro stimulated) when compared to the GI group (non-vaccinated and in vitro stimulated). The higher proliferation in CD3e^+^CD69^+^ cells indicates a greater early activation of the immune system following an in vitro SVA stimulation in the vaccinated group. Furthermore, vaccinated animals also demonstrated a greater proliferation of CD3e^+^CD4^+^CD44^+^CD62L^−^ cells, suggesting a recall of the effector T memory response. Together, these results show that vaccination can activate effector and peripheric memory cells. Although other cell subsets analyzed in this study did not have significant increases, the proliferation of these cell subpopulations demonstrates that vaccination triggers an adaptive and specific immune response, which is critical for the development of immunity. In previous studies on vaccines against SVA in mice, cellular immunity was evaluated only by cytokines [17] and a panel of T lymphocytes [20]. An increase in CD4^+^ and CD8^+^ T lymphocytes in mice vaccinated with a recombinant pseudorabies virus expressing SVA VP2 was observed [20]. Previous investigations using murine models did not examine additional markers, such as cellular activation and cellular memory, as we have analyzed in our study.

Despite the advantages associated with murine models, it is important to consider a few limitations in the use of animal models for the development of vaccines. These conditions include the lack of infectivity of the agent in the chosen animal model and the variations in the routes of infection [21]. Even after challenge with a high viral load (7 × 10^6^ to 2 × 10^7^ TCID_50_/mouse) of SVA via subcutaneous, intraperitoneal, and intramuscular routes, Balb/c and Kunming mice showed no signs of SVA infection [40]. Furthermore, variations among the viral isolates may exacerbate these limitations, compromising the outcomes. Here, the choice of the vaccine candidate was based exclusively on in silico analyses that involved phylogenetic inference and amino acid sequence alignment, resulting in the selection of a strain that covers 98% of the overall genetic diversity of the SVA circulating in the last few years in Brazil. The PARNAS algorithm was used to select a strain that best represents the observed genetic diversity of SVA in Brazil. The algorithm was previously applied to the influenza virus [41], but we demonstrated that it could be suitable for SVA as well. The selection of high pathogenicity strains of SVA can be considered one of the challenges in the production of SVA vaccines. According to Liu et al. (2020), even viruses isolated from vesicular fluid may not reproduce clinical disease after challenge in pigs [40]. Here, no assessment of viral infectivity and/or pathogenicity was performed. This might have contributed, even partially, to the low infection rate observed in mice. However, it is important to highlight that the virus used for the challenge was isolated from pigs during an outbreak of vesicular disease in a swine production facility. Furthermore, transposing the results from the animal model to the target species is a challenge. It is noteworthy that even in studies where neutralizing antibody titers were detected post-vaccination in mice, these titers are significantly lower than those observed when the vaccine is directly tested in pigs, the target species for SVA. This disparity suggests that, although these results may indicate protection in animal models, the validation of efficacy still needs to be evaluated in the target species.

In addition to the animal model, the evaluation of the inactivated vaccine formulations can be affected by other factors, including the antigenic load, inactivation process, and composition of adjuvants [42]. Here, the vaccine formulation included an SVA isolate with a viral titer of 10^7^ TCID_50_. Comparing different SVA titers of 10^5^, 10^6^, and 10^7^ TCID_50_ for the vaccine composition, it was demonstrated that the higher titer resulted in neutralizing antibody production and the absence of post-challenge viral shedding in Balb/c mice [19]. Regarding the vaccine’s manufacturing process, the viral inactivation was carried out with BEI for 8 h at a neutral pH. It was demonstrated that the inclusion of BEI at a final concentration of 1.5 mM at 37 °C resulted in an average viral reduction of one log/hour [30] while preserving pH neutrality. Picornaviruses are sensitive to changes in pH. A neutral environment maintains viral stability without releasing nucleic acid or destabilizing structural proteins, thereby preserving viral antigenicity [30,43]. This is critical because capsid proteins are the main antigens of inactivated vaccines, especially for viruses within the *Picornaviridae* family [43]. Because of the lack of viral replication, inactivated vaccines are often administered with an adjuvant to enhance viral-specific immune responses [35]. The W/O/W mineral adjuvant (MONTANIDE™ ISA 201 VG) chosen to compose the vaccine formulation induces short and long-term cellular and humoral immune responses and is compatible with inactivated antigens [44]. In the study conducted by Ibrahim et al. (2015), which compared various adjuvants in an inactivated FMD vaccine for cattle, they demonstrated an enhanced induction of a prompt cellular response (observed 14 days post-vaccination) with the W/O/W adjuvant [45]. This adjuvant was also selected for our study and was the most used in other SVA vaccine evaluations [11,13,16,17,19].

## 5. Conclusions

In summary, the results suggest that the inactivated vaccine against SVA, formulated with a representative contemporary Brazilian isolate, proved to be both safe and effective and capable of eliciting humoral and cellular immune responses against SVA in BALB/cByJ mice. This was evidenced by the presence of specific IgG antibodies and the activation of specific immune cells. However, the analysis of vaccine efficacy in the murine model was not possible as the detection of viral RNA in the organs occurred only in two non-vaccinated and challenged animals (GII), combined with a lower viral shedding in feces following an in vivo challenge, and cytometry results suggest that the majority of mice were resistant to SVA infection. The vaccine developed in this study will guide its application in pigs, playing a crucial role in the prevention and control of SVA infection.

## Figures and Tables

**Figure 1 vaccines-12-00845-f001:**
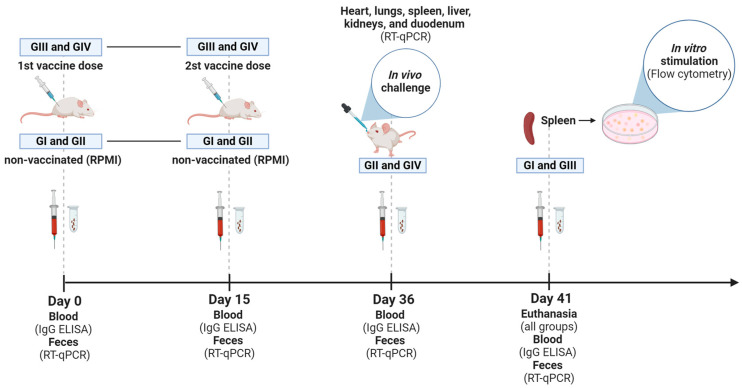
Experimental design. Timeline of GI (non-vaccinated and in vitro stimulated), GII (non-vaccinated and challenged), GIII (vaccinated and in vitro stimulated), and GIV (vaccinated and challenged) groups, highlighting the time points (Days 0, 15, 36, and 41) of blood and feces collection, administration of inactivated *Senecavirus A* vaccine (first and second doses) and euthanasia of mice. Illustration created with BioRender.com (accessed on 28 June 2024).

**Figure 2 vaccines-12-00845-f002:**
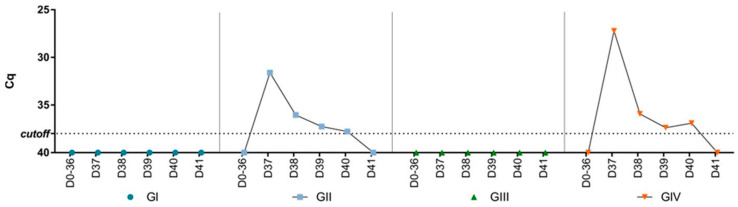
Cq values of SVA RNA load by RT-PCR in pooled fecal samples collected from mice before (days 0–36) and after (days 37–41) the challenge for the four groups (GI—non-vaccinated, GII—non-vaccinated and challenged, GIII—vaccinated, and GIV—vaccinated and challenged).

**Figure 3 vaccines-12-00845-f003:**
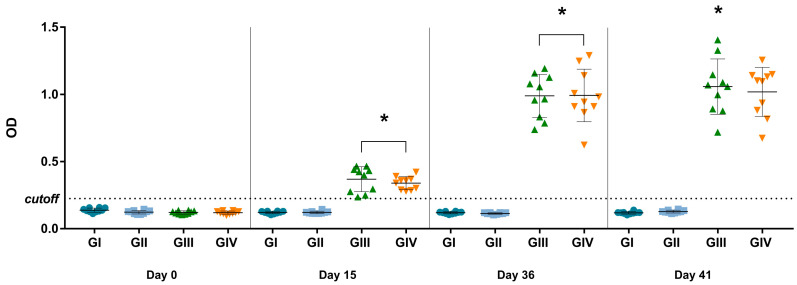
ELISA for detection of IgG against SVA at 0, 15, 36, and 41 dpv. The values are presented individually for each animal in its respective treatment group (GI—non-vaccinated; GII—non-vaccinated and challenged; GIII—vaccinated; GIV—vaccinated and challenged. The mean and ± standard deviation are represented by the black line ahead of the values. * *p* values ≤ 0.05 were considered statistically significant.

**Figure 4 vaccines-12-00845-f004:**
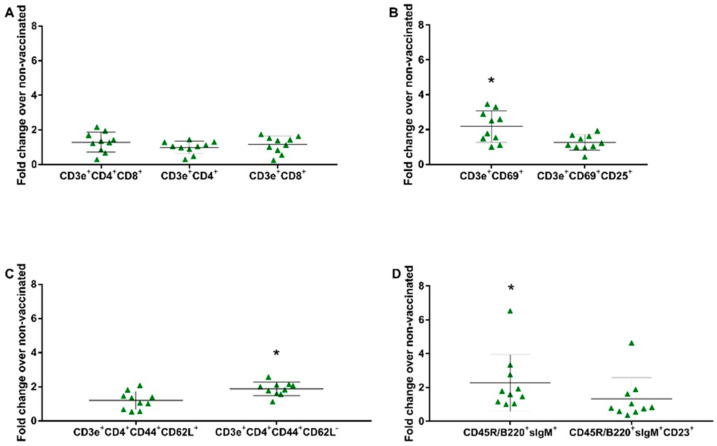
In vitro stimulation assay. Immune cells in the splenocytes proliferation assay stimulated with the vaccine virus (SVA/BRA/GO/946-22/2019—8000 TCID_50_) were compared as a fold change from the vaccinated group (GIII) over the non-vaccinated group (GI) at 41 dpv. (**A**) T Lymphocyte Subset (CD3e^+^CD4^+^; CD3e^+^CD8α^+^); (**B**) T Lymphocyte Activation (CD3^+^CD69^+^; CD3^+^CD69^+^CD25^+^); (**C**) Naïve/Memory T cell (CD3^+^CD4^+^CD44^+^CD62L^+^; CD3^+^CD4^+^CD44^+^CD62L^−^); (**D**) B Lymphocyte Subset (CD45R/B220^+^IgM^+^; CD45R/B220^+^IgM^+^CD23^+^). Data shown are the fold increase in the mean and ± standard deviation represented by the black line ahead of the values. * *p* values ≤ 0.05 were considered statistically significant.

**Table 1 vaccines-12-00845-t001:** Experimental groups, treatments, and challenges performed for the immunogenicity assessment of the inactivated vaccine against SVA in Balb/cByJ.

Experimental Groups	Treatments	
G I	Non-vaccinated (RPMI, IM *)	in vitro stimulation ^a^
G II	Non-vaccinated (RPMI, IM *)	challenge ^b^
G III	Vaccinated (10^7^ TCID_50_/100 µL + ISA 201 VG, IM *)	in vitro stimulation ^a^
G IV	Vaccinated (10^7^ TCID_50_/100 µL + ISA 201 VG, IM *)	challenge ^b^

^a^ SVA (SVA/BRA/GO/946-22/2019) 8000 TCID_50_ in mouse splenocyte cell culture. ^b^ SVA (SVA/BRA/GO/946-22/2019) titer 2 × 10^7^ TCID_50_/100 µL, 100 µL/mouse orally, 36 days post-vaccination (dpv). * IM: intramuscular route.

**Table 2 vaccines-12-00845-t002:** Panel of fluorochrome-labeled monoclonal antibodies.

Panel	Antibody	Fluorochrome
Mouse Naïve/Memory T cell (cod 561609, BD Becton Dickinson)	anti-CD44 anti-CD4 anti-CD62L anti-CD3e	PE PerCP-Cy™5.5 APC APC-Cy™7
Mouse T Lymphocyte Activation Antibody Cocktail (cod 557908, BD Becton Dickinson)	anti-CD25 anti-CD69 anti-CD3e	PE-Cy™7 PE APC
Mouse T Lymphocyte Subset Antibody Cocktail (cod 558431, BD Becton Dickinson)	anti-CD3e anti-CD4 anti-CD8α	PE-Cy™7 PE APC
Mouse B Lymphocyte Subset Antibody Cocktail (cod 558332, BD Becton Dickinson)	anti-CD45R/B220 anti-CD23 (FcεRII) anti-sIgM	PE-Cy™7 PE APC

## Data Availability

The data that support this study are available from the corresponding author upon reasonable request. Source data are provided in this paper.

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
