# Peer review of "Immunogenicity of an Inactivated Senecavirus A Vaccine with a Contemporary Brazilian Strain in Mice"

_vaccines, 2024, doi:10.3390/vaccines12080845_

Round 1

Reviewer 1 Report

Comments and Suggestions for Authors

The manuscript prepared by Amanda de Oliveira Barbosa et al., titled "Immunogenicity of an inactivated Senecavirus A vaccine with a contemporary Brazilian strain in mice," focuses on the evaluation of the immune efficacy of an inactivated vaccine using a Brazilian Senecavirus A epidemic strain. However, there are significant issues in the presentation of the experimental results. The following suggestions for modifications are provided:

1. In the Methods section, consider using a different term instead of "vitro challenge" to clarify that it refers to antigen stimulation outside the body.

2. There are ambiguities in the wording in lines 120-121 regarding the acquisition of "fecal samples" and "blood samples." Should "NB3" in line 126 be corrected to "BSL-3"? Change "Days 14" to "Days 15" in line 141.

3. The sequence analysis results of 50 Senecavirus A isolates in Results 3.1 are overly simplistic. Consider providing more detailed information such as alignment of important antigen genes, evolutionary distances, and variations.

4. The safety assessment of the vaccine is too simplistic. At least, data on weight changes should be included.

5. In Results 3.3, the quantitative analysis of "viral shedding" in feces seems to be based on data from only one sample, which is insufficient. Additionally, it is recommended to present data on viral loads in tissues using graphs. For nucleic acid quantification, absolute quantification methods are preferred over using only "Cq" values.

6. Present tissue slice images in Results 3.4.

7. In Results 3.5, it is suggested to include antibody testing against a distantly related strain concurrently, preferably conducting a virus neutralization test.

8. Provide the raw data for "flow cytometry" to clearly display the data for each marker.

Comments on the Quality of English Language

Minor editing of English language required.

Author Response

Comments 1: In the Methods section, consider using a different term instead of "vitro challenge" to clarify that it refers to antigen stimulation outside the body.

Response 1: The term “in vitro challenge” has been replaced for “in vitro stimulation” throughout the manuscript. Additionally, the term “in vivo challenge” has been replaced for “challenge” throughout the manuscript.

Comments 2: There are ambiguities in the wording in lines 120-121 regarding the acquisition of "fecal samples" and "blood samples." Should "NB3" in line 126 be corrected to "BSL-3"? Change "Days 14" to "Days 15" in line 141.

Response 2: The four experimental groups contained 10 mice each. Feces were collected from grouped-housed mice. For clarification, we modified the sentence in lines 134-137 as follows: “Fecal samples were collected from grouped-housed mice. For this, mice were transferred to another box and fresh feces were obtained. Blood samples were collected individually through retro-orbital puncture under sedation. Feces and blood were taken for all groups on days 0, 15, 36, and 41 after immunization or RPMI administration”.

The abbreviation “NB3” was replaced by “BSL-3” in line 140.

The information “Days 14” was replaced by “Days 15” in line 155.

Comments 3: The sequence analysis results of 50 Senecavirus A isolates in Results 3.1 are overly simplistic. Consider providing more detailed information such as alignment of important antigen genes, evolutionary distances, and variations.

Response 3: For a better understanding about the selection of the vaccine strain, we have included in section 2.2 (line 89), the following paragraph: “For this purpose, a dataset contained the 50 Brazilian sequences generated in this study, along with all SVA genome sequences available in GenBank (NCBI) up to 2022, was built. Nucleotide alignment was generated using MAFFT v7.490 [25], and manually corrected with Aliview v1.28 [26]. Final multiple sequence alignment in-cluded the coding region (~ 6546 in length) of 328 SVA collected from pigs globally, in-cluding 64 sequences from Brazil. To infer the evolutionary relationships for this da-taset, a maximum likelihood (ML) phylogenetic tree was reconstructed using the IQTREE v2.1.2 following the automatic best-fit model selection process (GTR+F+I+G4). The statistical support for branches within the inferred tree was assessed using SH-like approximate Likelihood-Ratio Test (SH-aLRT) [27] and Ultrafast Bootstrap approximation (UFBoot) approximation with 1,000 pseudoreplicates [28,29]. Based on this global phylogenetic inference (Figure S1 in the supplementary material), the more recent sequences of SVA strains isolated in Brazil (2018-2021) (n = 51) were used to infer a separately ML tree using the settings described previously, and a multiple amino acid alignment (ORF1, 2181 aa) was generated using MAFFT. PARNAS algorithm was executed with the option to rescale the tree edges with the number of ORF1 amino acid substitutions and with 2% of amino acid divergence between SVA sequences. The pairwise amino acid identity matrix was generated using SDT v1.2 (Figure S2 in the supplementary material).”

We included in section 3.1 (line 249) of the Results the following information: “The percent of amino acid identity among the 51 Brazilian SVA isolates was 98.2% or higher. The sequence chosen by PARNAS as the best representative showed an overall amino acid identity ranging from 98.67% to 99.82% compared to the other sequences generated in this study”.

Comments 4: The safety assessment of the vaccine is too simplistic. At least, data on weight changes should be included.

Response 4: We do agree with the reviewer´s comment and suggestion. Unfortunately, data regarding weight changes were not recorded during the experiment. However, we did not observe any changes in the water and food intake. In addition, we only have checked for changes in their behaviour pattern and major adverse reactions after vaccination. It is also important to mention that safety regarding the use of BEI for viral inactivation and Montanide ISA 201 as an adjuvant, are well documented in the literature.

Comments 5: In Results 3.3, the quantitative analysis of "viral shedding" in feces seems to be based on data from only one sample, which is insufficient. Additionally, it is recommended to present data on viral loads in tissues using graphs. For nucleic acid quantification, absolute quantification methods are preferred over using only "Cq" values.

Response 5: The value concerning viral excretion in feces presented in item 3.3 of the Results is derived from pooled fecal samples (i.e., one pooled sample per group). We opted for pooled collection of feces as the animals did not have individual identification, so it would not be possible to correlate the excretion loads individually on the evaluated days. We made minor changes on the legend of figure 1 (line 274), to emphasize that we used pooled feces, and each analyzed sample represented 10 mice of each group.

Regarding the viral load in the organs after the challenge, unfortunately, we did not perform absolute quantification. However, we standardized the weights of the organs used in the extraction to establish a comparison between the vaccinated and non-vaccinated groups based on the Cq values obtained in the RT-PCR reaction carried out according to Joshi et al. (2016).

Comments 6: Present tissue slice images in Results 3.4.

Response 6: We appreciate the reviewer's suggestion. Since no specific microscopic lesions were observed in any of the analyzed tissues, we chose not to include the images in the manuscript. However, we can provide a figure comprising the non-specific lesions observed in the heart, kidney, liver, and duodenum, if necessary.

Comments 7: In Results 3.5, it is suggested to include antibody testing against a distantly related strain concurrently, preferably conducting a virus neutralization test.

Response 7: We do agree with the reviewer that including cross-neutralization results would be valuable for the manuscript. However, our attempts to perform sero-neutralization assays using the homologous virus were unsuccessful. As noted in the discussion (lines 361-363), the serum from both vaccinated and non-vaccinated animals proved to be toxic to H1299 and BHK-21 cells. Also, based on the analysis of aminoacid sequences there was a high identity (98.2-99.9%) among the virus strains used for the selection of the vaccine strain.

Comments 8: Provide the raw data for "flow cytometry" to clearly display the data for each marker.

Response 8: A new figure, showing the hierarchical gating strategy applied to samples from the experiment, was added in the supplementary material (Figure S4).

Reviewer 2 Report

Comments and Suggestions for Authors

Review report

The MS describe data on immunological profile of mice vaccinated with an inactivated SVA vaccine prepared from contemporary Brazilian SVA isolate. Seneca virus A is an emerging viral pathogen in swine with significant economic implications due to direct production losses, increased costs of disease management, and potential trade restrictions. Therefore, efforts to come up with effective control and mitigation strategies are essential to minimize these economic impacts and ensure the sustainability of the swine industry. The study used mice which are not natural host for SVA, hence their immune response may differ significantly from that of pigs, thus, the findings should be validated in pigs to ensure they are relevant to the target species. Having said that, there are major issues that need to be addressed;

11.  Overall, these findings suggest that SVA did not establish a productive infection in the majority of mice used in our study (L360-361). With this conclusion it means that the there is no correlation between immune responses reported and SVA infection.

22. Moreover, the conclusion that “the results suggest that the inactivated vaccine against SVA is effective is not well supported by the results. E.g.,

a.      At 1 and 4 dpc, viral shedding was higher in the vaccinated group (GIV) when compared to the non-vaccinated group (GII) (L247-248). The viral shedding to be higher in control than the vaccinated, what could be the reason for the reverse observation?

b.     No lesion suggestive of SVA infection was observed in both vaccinated and non-vaccinated groups (L260).

33.  The adjuvant used in this study have ability to induce short and long-term cellular and humoral immune responses, however, the study design did not account for the adjuvant effects. There was no group for adjuvant only or RPMI+adjuvant to ensure that the immune response observed in vaccinated group is not due to adjuvant.

44.  “an anamnestic serological response triggered by the inactivated vaccine was observed (L326-327)”. Looking at Figure 3, this statement is not supported, since there is no difference in IgG levels of the vaccinated groups at day 36 (prior to challenge) and day 41 (after challenge).

Other questions

i. Why didn’t the authors report IgG Ab titers rather than OD values?

ii. Why was neutralization Ab assay not done?

iii. Why didn’t the authors examined the frequencies of different T cell populations in spleen MNCs before carrying out invitro cell proliferation assay?

Minor comments

L119 –  “(100 μL/dose, IM)” give exact viral concentration

L131 – provide details how euthanasia was done with specific drug and doses (mg/Kg body weight)

L139 - The flow chat is difficult to follow. There should be clear separation of vaccinated and mock groups in Day 0 and 15.

L141 – “14” should be replaced with “15” as per flow chart.

L209 – write FMO in full when used first time.

L236 – Provide supplementary figures to support this (Agar plate, or broth and cell line plate).

L260 - Reference images of the staining are needed as supplementary figures.

L267 – How did the authors determine the cutoff OD? Can they report Ab response in IgG titers rather than OD values?

L275 - positive of what? However, I expected to see some Ab response (though low) from non-vaccinated group after challenge (Day 41).

L372 – “data not shown” - Can this data be shown as supplementary to benefit the readers?

Comments on the Quality of English Language

The English language is sufficient

Author Response

Comments 1: Overall, these findings suggest that SVA did not establish a productive infection in the majority of mice used in our study (L360-361). With this conclusion it means that the there is no correlation between immune responses reported and SVA infection.

Response 1: The study initially categorized participants into four groups: the non-vaccinated and non-challenged group (negative control), the vaccinated and non-challenged group, the non-vaccinated and challenged group (positive control), and the vaccinated and challenged group. However, the positive control group failed to elicit an adequate response or produce an infection, leading to the exclusion of both challenged groups from further evaluation of the cellular response. Consequently, the immune responses discussed in the study pertain solely to the non-challenged groups.

Comments 2: Moreover, the conclusion that “the results suggest that the inactivated vaccine against SVA is effective is not well supported by the results. E.g.,

  1. At 1 and 4 dpc, viral shedding was higher in the vaccinated group (GIV) when compared to the non-vaccinated group (GII) (L247-248). The viral shedding to be higher in control than the vaccinated, what could be the reason for the reverse observation?
  2. No lesion suggestive of SVA infection was observed in both vaccinated and non-vaccinated groups (L260).

Response 2: The challenge test is crucial for demonstrating vaccine efficacy; however, it is also possible to establish the vaccine's immunogenicity without such a test. Based on this understanding, along with several publications, including prior works by authors of this article (e.g., PMID: 37587490 and PMID: 37605141), the conclusion of the work is based mainly on the results found on the immunogenicity of the vaccine. The findings from comparisons between the vaccinated and non-vaccinated groups clearly illustrate the differences in humoral response analyses and cellular lymphoproliferation assays, forming the basis of the study's conclusions.

Additionally, as described in lines 463-465, the authors conclude that “the results suggest that the inactivated vaccine against SVA, formulated with a representative contemporary Brazilian isolate, proved to be both safe and effective in eliciting humoral and cellular immune responses against SVA in BALB/cByJ mice.”

Also, the results referenced by the reviewer indicating viral shedding in GII and GIV and the absence of lesions typical of SVA infection led us to conclude that the infection was not successfully reproduced in the challenged animals. The higher viral shedding observed in animals from vaccinated group (GIV) could be due to individual variation. We incorporated these findings into the discussion section of our work. Therefore, we were unable to conduct thorough analyses of vaccine efficacy. However, this limitation did not hinder our assessment of immunogenicity. Additionally, this study utilized a murine model for optimizing dosages, challenges, and vaccine administration, among other variables. This model played a crucial role in minimizing future animal use and experimental costs. Nevertheless, conducting clinical trials in the target species remains essential.

Comments 3: The adjuvant used in this study have ability to induce short and long-term cellular and humoral immune responses, however, the study design did not account for the adjuvant effects. There was no group for adjuvant only or RPMI+adjuvant to ensure that the immune response observed in vaccinated group is not due to adjuvant.

Response 3: That is an excellent question. We can conclude that the observed humoral response is attributable to the vaccination (inoculum plus adjuvant) and not merely the adjuvant, because the measurement was specific to IgG antibodies targeting SVA, rather than total IgG levels. Similarly, the cellular immune response was assessed using the cell proliferation marker CFSE, which quantifies the marker's intensity following specific exposure to the vaccine virus.

It would be challenging to distinguish between the effects of the vaccine and the adjuvant if we measured only the total concentrations of IgG and cytokines, or if we analyzed the cellular response without conducting stimulation and proliferation assays.

Comments 4: “an anamnestic serological response triggered by the inactivated vaccine was observed (L326-327)”. Looking at Figure 3, this statement is not supported, since there is no difference in IgG levels of the vaccinated groups at day 36 (prior to challenge) and day 41 (after challenge).

Response 4: Indeed, these findings also led us to conclude that the challenge did not induce infection in the animals, suggesting that the virus selected for the challenge may exhibit low pathogenicity in mice. This observation was also documented and discussed in the fifth paragraph of the discussion section. The authors refer to the “anamnestic” humoral immune response elicited by vaccination, which can be observed on days 15 and 36, before any interference from the in vivo challenge.

Comments 5: Why didn’t the authors report IgG Ab titers rather than OD values?

Response 5: We have measured the levels of specific IgGs using an in-house ELISA and no SVA-positive mouse sera with known IgG titers were available for standardization. Therefore, it was not possible to convert the OD values into IgG antibody titers.

Comments 6: Why was neutralization Ab assay not done?

Response 6: We do agree with the reviewer that including sero-neutralization results would be valuable. However, our attempts to perform sero-neutralization assays were unsuccessful. As noted in the discussion (lines 361-363), the serum from both vaccinated and non-vaccinated animals proved toxic to H1299 and BHK-21 cells.

Comments 7: Why didn’t the authors examined the frequencies of different T cell populations in spleen MNCs before carrying out in vitro cell proliferation assay?

Response 7: Because the animals used are isogenic, the negative control group (not vaccinated and not challenged) serves as the baseline for these animals.

Comments 8: L119 –  “(100 μL/dose, IM)” give exact viral concentration

Response 8: The information “(107 TCID50/100 µL + ISA 201 VG, IM)” was added (line 132).

Comments 9: L131 – provide details how euthanasia was done with specific drug and doses (mg/Kg body weight)

Response 9: The information “all mice were euthanized with intraperitoneal ketamine (300 μg/g) and xylazine (40 μg/g body weight) injection.” was added (lines 146-147).

Comments 10: L139 - The flow chat is difficult to follow. There should be clear separation of vaccinated and mock groups in Day 0 and 15.

Response 10: We do agree with the reviewer. The flowchart was modified showing a clear separation of vaccinated and mock groups in day 0 and 15.

Comments 11: L141 – “14” should be replaced with “15” as per flow chart.

Response 11: The information “Days 14” was replaced by “Days 15” in line 155.

Comments 12: L209 – write FMO in full when used first time.

Response 12: The information “FMO” was replaced by “Fluorescence Minus One (FMO)” in line 223.

Comments 13: L236 – Provide supplementary figures to support this (Agar plate, or broth and cell line plate).

Response 13: A figure showing no bacterial growth on blood agar and MacConkey agar was added to the supplementary material (Figure S3).

Comments 14: L260 - Reference images of the staining are needed as supplementary figures.

Response 14: We appreciate the reviewer's suggestion. Since no specific microscopic lesions were observed in any of the analyzed tissues, we chose not to include the images in the manuscript. However, we can provide a figure comprising the non-specific lesions observed in the heart, kidney, liver, and duodenum, if necessary.

Comments 15: L267 – How did the authors determine the cutoff OD? Can they report Ab response in IgG titers rather than OD values?

Response 15: The cutoff was calculated following the authors Frey, Di Canzio, and Zurakowski (1998), as described on line 191.  Regarding the conversion of the OD values into IgG titers, we measured the levels of specific IgGs using an in-house ELISA, and we did not have SVA-positive mouse sera with known concentrations of IgG titers. Therefore, it was not possible to convert the OD values into IgG antibody titers.

Comments 16: L275 - positive of what? However, I expected to see some Ab response (though low) from non-vaccinated group after challenge (Day 41).

Response 16: The paragraph was modified to: "None of the mice from the non-vaccinated groups (GI and GII) tested positive for SVA during the experimental days according to the in-house IgG ELISA." (lines 293-294). Regarding the expected antibody response from vaccinated group after challenge, we agree with the reviewer. However, we evaluated IgG levels only 5 days after challenge and all animals were euthanized at day 41. Thus, we cannot conclude that the mice didn’t produce IgG due to the short period of evaluation.

Comments 17: L372 – “data not shown” - Can this data be shown as supplementary to benefit the readers?

Response 17: We would like to thank the reviewer for the comment. However, we did not observe significative differences between experimental groups. Thus, data regarding IgM levels were not relevant for the main conclusions of the study.

Reviewer 3 Report

Comments and Suggestions for Authors

This study performed a thorough analysis of a contemporary Senecavirus A strain as inactivated vaccine in a murine model. 50 virus strains were gathered, and one was selected for vaccine formulation for the analysis of immune response and protectivity in mice. Cellular immune response was also analyzed in depth with surface markers. Overall, the study provided ample content and useful data, as much as what can be gathered from a murine model.

There are some minor points:

I believe it is misleading to use the term in vitro ‘challenge’, and instead, ‘stimulation’ should be used. The word ‘challenge’ is usually reserved for experiments that involve testing the ability to neutralize the virus/pathogen. In this study, the application of the virus to the splenocytes is to observe cell response and proliferation, and therefore ‘stimulation’ would be more commonly used.

Also, based on the above point, the presentation of the experimental design can be simplified to GII and GIV.

It would have been interesting to include a historical SVA strain in the study for comparison purposes.

Comments on the Quality of English Language

Only minor errors detected. For example, Line 344 might need some correction: 'it was not ruled out the presence of neutralizing antibodies in vaccinated animals.'

Author Response

Comments 1: I believe it is misleading to use the term in vitro ‘challenge’, and instead, ‘stimulation’ should be used. The word ‘challenge’ is usually reserved for experiments that involve testing the ability to neutralize the virus/pathogen. In this study, the application of the virus to the splenocytes is to observe cell response and proliferation, and therefore ‘stimulation’ would be more commonly used.

Response 1: The term “in vitro challenge” has been replaced with “in vitro stimulation” throughout the manuscript and also in figure 1 of the Experimental Design.

Comments 2: Also, based on the above point, the presentation of the experimental design can be simplified to GII and GIV.

Response 2: We appreciate the reviewer's feedback on the manuscript.  However, to provide a comprehensive overview and a better understanding of the experiment, we would like to retain the original presentation of the experimental design including all four groups.

Comments 3: It would have been interesting to include a historical SVA strain in the study for comparison purposes.

Response 3: We do agree with the reviewer that including cross-neutralization results would be valuable for the manuscript. However, our attempts to perform sero-neutralization assays using the homologous virus were unsuccessful. As noted in the discussion (lines 361-363), the serum from both vaccinated and non-vaccinated animals proved toxic to H1299 and BHK-21 cell cultures. Also, based on the analysis of aminoacid sequences there was a high identity (98.2-99.9%) among the virus strains used for the selection of the vaccine strain.

Round 2

Reviewer 1 Report

Comments and Suggestions for Authors The authors have responded comprehensively to the initial round of revision comments; however, upon reevaluation of the manuscript, several critical concerns persist regarding the suitability of mice as an effective model for evaluating the immunogenicity of the Senecavirus A vaccine. Firstly, the study fails to convincingly establish mice as a robust animal model for assessing the vaccine's immune protective efficacy against Senecavirus A. This undermines the broader significance of the findings. Secondly, the manuscript's results on humoral and cellular immune responses exhibits substantial deficiencies, inadequately showcasing the vaccine's immunological effectiveness. Therefore, while the authors have made efforts to address previous feedback, the manuscript still requires substantial revisions to strengthen the interpretation and significance of the results obtained. Comments on the Quality of English Language

 Minor editing of English language required.

Author Response

Comment: The authors have responded comprehensively to the initial round of revision comments; however, upon reevaluation of the manuscript, several critical concerns persist regarding the suitability of mice as an effective model for evaluating the immunogenicity of the Senecavirus A vaccine. Firstly, the study fails to convincingly establish mice as a robust animal model for assessing the vaccine's immune protective efficacy against Senecavirus A. This undermines the broader significance of the findings. Secondly, the manuscript's results on humoral and cellular immune responses exhibit substantial deficiencies, inadequately showcasing the vaccine's immunological effectiveness. Therefore, while the authors have made efforts to address previous feedback, the manuscript still requires substantial revisions to strengthen the interpretation and significance of the results obtained.

Response: The analysis of vaccine efficacy and immunogenicity are two critical aspects of vaccine development and evaluation, each focusing on different areas of assessment. In evaluating vaccine efficacy, the primary objective is to determine how much the vaccine reduces the incidence of the disease compared to an unvaccinated group. In contrast, immunogenicity refers to the ability of a vaccine to induce an immune response in the organism, such as the production of antibodies or the activation of specific immune cells.

The initial proposal of our study, with an experimental design, was to analyze both the immunogenicity and efficacy of the vaccine in the murine model. However, the viral challenge was not suitable for this animal model, as the positive control (unchallenged + challenged) did not present data supporting viral replication and infection. Despite this, we consider it important to share these findings with the scientific community, as there are studies that have demonstrated SVA infection in the murine model (doi:10.1016/j.rvsc.2021.12.010; doi:10.1128/spectrum.05229-22). Nevertheless, data from the vaccinated and unvaccinated groups (both unchallenged) showed significant differences in immune response, such as specific SVA antibodies and proliferated (reactive) T cell activity specifically post-SVA stimulation, demonstrating the vaccine's immunogenicity.

We emphasize that both vaccine efficacy analysis, which measures the reduction of disease cases, and immunogenicity, which measures the immune response generated by the vaccine, are essential analyses to ensure that the vaccine can induce an immune response (immunogenicity) and that this response is effective in preventing the disease in pigs (vaccine efficacy). The use of the murine model aimed to reduce experimental costs and optimize the vaccine, adjuvant, and dose before proceeding to pigs was important; however, we will need to conduct clinical trials in pigs to analyze vaccine efficacy, which is part of the progression in the development of any vaccine.

In order to make it clear that the data obtained by the study allowed the assessment of the vaccine immunogenicity, we included a sentence in the discussion section on line 382, as follows: The absence of a productive infection in the challenged mice prevented us from assessing the vaccine's ability to reduce infection or prevent disease in the murine model, thereby hindering the analysis of vaccine effectiveness. Nonetheless, the generation of antibodies SVA-specific enabled us to evaluate the vaccine's immunogenicity”.

Also, the conclusion on line 466 was modified as follows: “In summary, the results suggest that the inactivated vaccine against SVA, formulated with a representative contemporary Brazilian isolate, proved to be both safe and capable of eliciting humoral and cellular immune responses against SVA in BALB/cByJ mice. This was evidenced by the presence of specific IgG antibodies and the activation of specific immune cells. However, the analysis of vaccine efficacy in the murine model was not possible, as the detection of viral RNA in the organs occurred only in two non-vaccinated and challenged animals (GII), combined with a lower viral shedding in feces following in vivo challenge, and cytometry results, suggesting that the majority of mice were resistant to SVA infection”.

Reviewer 2 Report

Comments and Suggestions for Authors

The authors have responded to my comments adequately.

Minor comment

Comments 15: L267 – How did the authors determine the cutoff OD? Can they report Ab response in IgG titers rather than OD values?

Response 15: The cutoff was calculated following the authors Frey, Di Canzio, and Zurakowski (1998), as described on line 191.  Regarding the conversion of the OD values into IgG titers, we measured the levels of specific IgGs using an in-house ELISA, and we did not have SVA-positive mouse sera with known concentrations of IgG titers. Therefore, it was not possible to convert the OD values into IgG antibody titers.

Comment: Although many studies have reported the Ab responses using OD values, the authors response about calculating Ab titers cutoff using SVA positive serum is not convincing, since cutoff in most ELISAs is calculated using known negative control serum (Average of negative control OD +3SD of negative control OD). Reporting Ab titers could have showed the sensitivity of the in-house ELISA used.

Comments 10: L139 - The flow chat is difficult to follow. There should be clear separation of vaccinated and mock groups in Day 0 and 15.

Response 10: We do agree with the reviewer. The flowchart was modified showing a clear separation of vaccinated and mock groups in day 0 and 15.

Comment: The flow chart still need modification. GII is non-vax and is grouped with Vax, similarly GIII is vax and is grouped with non-vax.  

Comments on the Quality of English Language

English quality good

Author Response

Reviewer # 2

Comment: Although many studies have reported the Ab responses using OD values, the authors response about calculating Ab titers cutoff using SVA positive serum is not convincing, since cutoff in most ELISAs is calculated using known negative control serum (Average of negative control OD +3SD of negative control OD). Reporting Ab titers could have showed the sensitivity of the in-house ELISA used.

Response: To establish the cutoff for the ELISA IgG for SVA antibodies used in the study, we utilized SVA negative sera from non-vaccinated groups (GI and GII). The cutoff was calculated using the formula described by Frey, Di Canzio, and Zurakowski (1998). This resulted in a cutoff value of 0.225. Alternatively, calculating the cutoff value using the mean optical density of the negative control OD plus three standard deviations (3SD) yielded a value of 0.280. Considering the minimal difference between these two values, we believe that there is no need of modify the cutoff since no difference will be observed in the results.

Comment: The flow chart still need modification. GII is non-vax and is grouped with Vax, similarly GIII is vax and is grouped with non-vax. 

Response: The flowchart was modified to clearly delineate the vaccinated and mock groups according to the experimental design throughout the entire study period.

Round 3

Reviewer 1 Report

Comments and Suggestions for Authors

The manuscript provides limited novel insights into Senecavirus A inactivated vaccine development. The formulation lacks innovation, and crucially, there is no assessment of the neutralizing antibody titers post-immunization. Furthermore, the manuscript fails to evaluate differences in neutralization efficacy against various viral variants. These aspects significantly limit the manuscript's contribution to the field of vaccine research against Senecavirus A.

Comments on the Quality of English Language

Minor editing of English language required

Author Response

Comment 1: The manuscript provides limited novel insights into Senecavirus A inactivated vaccine development. The formulation lacks innovation, and crucially, there is no assessment of the neutralizing antibody titers post-immunization. Furthermore, the manuscript fails to evaluate differences in neutralization efficacy against various viral variants. These aspects significantly limit the manuscript's contribution to the field of vaccine research against Senecavirus A.

Response 1: We thank the reviewer for the opportunity to address your comments and provide additional clarifications. Minor English language editing and grammar revisions were made along the manuscript.

In this study, we developed and evaluated the immunogenicity of an inactivated vaccine against Senecavirus A. Although inactivated vaccines are not a novel approach, our study employed the PARNAS algorithm to select a representative SVA brazilian contemporary strain for the vaccine formulation. In addition, we provide data of full-genome sequencing of 50 Brazilian SVA sequences. To our knowledge, no published studies on SVA vaccines have utilized such a rational selection process for a vaccine candidate. Furthermore, there are currently very few commercial vaccines available for Senecavirus (SVA) globally. In many endemic regions, countries rely on autogenous vaccines due to the absence of commercial options, making our vaccine particularly noteworthy. Additionally, our study provides novel data on cellular immune responses, an area that has not yet been explored in studies involving inactivated vaccines against SVA.

Regarding the neutralizing antibody titers, despite our efforts, the seroneutralization assay did not yield viable results for Senecavirus A due to toxicity observed in two different cell lines (H1299 and BHK) by the serum samples. As neutralization is essential to test the reactivity to homologous and heterologous viruses, this evaluation could not be performed. However, we conducted an ELISA assay using the homologous virus to provide a profile of the humoral immune response following vaccination. Recognizing the importance of this data, we are addressing this issue using pigs in an ongoing experiment evaluating the immunogenicity and cross-protection of the vaccine candidate strain.

We appreciate the thorough review of our manuscript. Despite the limitations of the study in mice, we believe that our data provide a valuable scientific contribution to the advancement of senecavirus vaccinology research.